# Botulinum Neurotoxin Type a Injection Combined with Absorbable Punctal Plug Insertion: An Effective Therapy for Blepharospasm Patients with Dry Eye

**DOI:** 10.3390/jcm12030877

**Published:** 2023-01-22

**Authors:** Malachie Ndikumukiza, Yu-Ting Xiao, You-Fan Ye, Jia-Song Wang, Xi Peng, Hua-Tao Xie, Ming-Chang Zhang

**Affiliations:** Department of Ophthalmology, Union Hospital, Tongji Medical College, Huazhong University of Science and Technology, Wuhan 430022, China

**Keywords:** absorbable punctum plug, botulinum neurotoxin type A, blepharospasm, dry eye

## Abstract

Blepharospasm patients often have dry eye manifestations. Botulinum neurotoxin type A (BoNT-A) injection has been the main management for blepharospasm and absorbable punctal plug (APP) insertion is shown to be effective in the treatment of dry eye. However, there have been no studies investigating the combined treatment of BoNT-A and APP in blepharospasm patients with dry eye. In this retrospective study, 17 blepharospasm patients with dry eye treated by BoNT-A injection and 12 receiving BoNT-A plus APP treatment were enrolled. The efficacy was evaluated according to the Jankovic rating scale, Ocular Surface Disease Index (OSDI), fluorescein staining (FL), fluorescein tear break-up time (FBUT) and Schirmer I test (SIT). Both BoNT-A and BoNT-A+APP treatment effectively reduced the functional impairment of blepharospasm. At baseline, all the patients had high OSDI scores (BoNT-A group: 82.48 ± 7.37, BoNT-A+APP group: 78.82 ± 4.60, *p* = 0.112), but relatively low degrees of FL (BoNT-A group: 3.18 ± 1.01, BoNT-A+APP group: 3.50 ± 1.24, *p* = 0.466), FBUT (BoNT-A group: 1.71 ± 0.77, BoNT-A+APP group: 2.17 ± 0.58, *p* = 0.077) and SIT (BoNT-A group: 2.53 ± 0.99, BoNT-A+APP group: 3.17 ± 1.23, *p* = 0.153). After treatment, OSDI, FL, FBUT and SIT were all obviously restored in the two groups. When comparing the changing rates, only OSDI (BoNT-A group: −52.23% ± 15.57%, BoNT-A+APP group: −61.84% ± 9.10%, *p* = 0.047) and FL (BoNT-A group: −22.55% ± 25.98%, BoNT-A+APP group: −41.94% ± 14.46%, *p* = 0.016) showed significant differences between the two groups. This study suggests that OSDI is not applicable in the diagnosis of dry eye among blepharospasm patients. For blepharospasm patients with severe dry eye symptoms, especially those with fluorescein staining in the cornea, the combined treatment of BoNT-A and APP is more effective than using BoNT-A alone.

## 1. Introduction

Blepharospasm, a subtype of focal dystonia, is often manifested as the abnormal contraction of the orbicularis oculi muscle [1,2]. Benign essential blepharospasm (BEB) is focal dystonia defined by an involuntary movement, persistent, bilateral and dystonic contraction of the eyelid muscles, resulting in partial or complete eyelid closure [3,4,5]. Currently, blepharospasm is a clinical diagnosis. However, it remains poorly understood and low in diagnosis rate [6]. Blepharospasm does not interfere significantly with daily activities, but its severity can extend to severe cases that render individuals functionally blind, preventing them from working, driving, walking and reading [7]. The intramuscular injection of botulinum neurotoxin type A (BoNT-A) is the only treatment that has shown improvement in blepharospasm symptoms over the past three decades [8,9].

As reported, blepharospasm and dry eye commonly occur together [10,11]. Dry eye, a multifactorial disease of the ocular surface and tear, is often accompanied by potential damage to the ocular surface, tear film instability, eye discomfort and visual disturbance [12,13]. In the early stages, blepharospasm is often misdiagnosed as dry eye because its symptoms imitate those of dry eye disease at first glance [10]. Observational data have demonstrated that a multitude of blepharospasm patients also suffer from dry eye to different degrees, which is owed to the same token of the involuntary movement of eyelid retractors and protractors [10,14]. According to prior studies, the lower pretarsal orbicularis oculi injections of BoNT-A could help to alleviate dry eye symptoms [1,15]. For patients with dry eye, absorbable punctal plug (APP) occlusion treatment has shown an improvement in dry eye diseases that do not respond to ocular medications [16,17]. However, studies on the combining procedures of BoNT-A and APP in the treatment of blepharospasm patients with dry eye are still scarce.

This study aims to evaluate the clinical effectiveness of the combined treatment of BoNT-A injection and APP insertion on blepharospasm patients with dry eye. We compared the symptoms and signs associated with blepharospasm and dry eye in patients managed by BoNT-A+APP and in those treated by BoNT-A only. The changing rates of dry eye symptoms and signs before and after treatment were also investigated.

## 2. Materials and Methods

### 2.1. Study Design

This retrospective consecutive cohort study was conducted at Union Hospital, Huazhong University of Sciences and Technology, China. The patients were retrospectively recruited from 1 January 2020 to 31 May 2021. The study adhered to the tenets of the Declaration of Helsinki and was approved by the ethics committee of Union Hospital, Huazhong University of Sciences and Technology.

### 2.2. Eligibility

Twenty-nine adults diagnosed with blepharospasm with typical dry eye manifestations but failing to respond to ocular medications were consecutively enrolled. All the patients were administered the standardized dry eye treatment using artificial tears (sodium hyaluronate eye drops; Santen Pharmaceutical Co., Ltd., Osaka, Japan) three times a day before, but did not show any improvement in the dry eye symptoms. Twelve patients receiving the combined treatment of BoNT-A and APP were assigned to the observation group. Seventeen patients undergoing BoNT-A injection alone served as the control group. The diagnosis of blepharospasm was made in line with the published standard criteria [17]. Subjects with immune-compromised status, hemifacial spasm or other neurologic abnormalities were not eligible. Patients with ocular surface defects such as conjunctivitis or keratitis, or who did not accept BoNT-A injection were also eliminated.

### 2.3. Assessments

The symptoms of blepharospasm were clinically graded according to the Jankovic rating scale. This scale includes two subscales that measure the severity and frequency of eyelid spasms, in each case on a 5-point scale ranging from 0 to 4 [18].

A validated Chinese-translated version of the Ocular Surface Disease Index (OSDI) questionnaire was used for the assessment of dry eye symptoms. The overall OSDI score was determined as suggested in a previous study and defined the dry eye symptoms as mild (13–22), moderate (23–32) or severe (≥33) [19].

After completing the questionnaire, the participants underwent a rigorous ophthalmologic investigation by the same specialist of the anterior segment. Firstly, fluorescein staining (FL) was conducted after dropping 2% sterile fluorescein sodium staining solution into the conjunctival sac and the examination was conducted for 2–3 min counting from the last instillation by passing light through a cobalt-blue filter and examined via a slit lamp. The classification of the entire corneal surface was facilitated by lifting the upper eyelid. The severity of corneal punctate dots was recorded using Oxford Schema: 0–5 of staining ranges for each panel (cornea, temporal conjunctiva and nasal conjunctiva) and 0–15 for the entire exposed inter-palpebral conjunctiva and cornea [20].

In the detection of the fluorescein tear break-up time (FBUT), patients were asked not to blink after fluorescein instillation. The length of time after a patient blinked to the introduction of dryness in the tear film was calculated in seconds. The evaluation was repeated three times, and the mean value was enrolled [20].

We performed Schirmer I test (SIT) without anesthesia to quantify the production of tears. A filter paper (5 × 35 mm) was placed on the lower conjunctiva within the temporal one-third area. After 5 min, the length traveled by the tears on the filter paper was measured. SIT was indicative of a water-deficient state if the measured distance was less than 5mm [21].

All patients in this study were evaluated with Jankovic grading, OSDI, FL, FBUT and SIT before and after BoNT-A injections or BoNT-A+APP treatment. The changing rates ((after-before)/before) of each index were compared in the two groups.

### 2.4. Treatment and Follow-Up Procedures

The BoNT-A injection preparation was made by mixing 100 units of purified dry botulinum toxin (Botox; Allergan, Inc.; Madison, WI, USA) with a 2.5 mL of saline solution. In the periocular region, the injection of BoNT-A was performed subcutaneously at 17 selected sites and the dose was 2 or 4 units per small or big dot injection site (Figure 1). The standard injection procedure was performed in all patients. The duration until the first injection (interval between diagnosis and the visit at which the injections were made) for all patients was in the range of 1–7 days.

For patients receiving BoNT-A+APP combination treatment, APP (Dissolvable Visiplug, Lacrimedics Inc.; Eastsound, WA, USA) was used to occlude the upper and lower lacrimal puncta in both eyes after BoNT-A injection.

After BoNT-A or BoNT-A+APP management, there was no specific treatment in those patients except using sodium hyaluronate eye drops three times a day if the patient needed it. The patients were followed up regularly on the next day, the first week and 4–6 weeks after BoNT-A injection or BoNT-A+APP treatment, and the clinical features were recorded before treatment, one week after and at the last visit.

### 2.5. Statistics

For all variables in this study, the arithmetic mean ± standard deviation data were calculated. An independent-samples *t*-test was used for statistical comparisons between the BoNT-A+APP group and the BoNT-A group. A paired-samples *t*-test was applied to compare the efficacy before and after treatment in the same group. We considered a *p*-value inferior to 0.05 as significant statistical.

## 3. Results

### 3.1. Patient Characteristics

A total of 29 right eyes from 29 patients with blepharospasm and typical dry eye manifestations were included in the study. The demographic characteristics of the patients were almost the same among the two groups. In this study, 64.5% (11/17) of patients in the BoNT-A group and 75% (9/12) in the BoNT-A+APP group were female. Most of the patients were over 60 years old and the median ages were 67 (58.5–73.0) and 64 (59.3–67.0) years in the BoNT-A group and the BoNT-A+APP group (Table 1), respectively.

### 3.2. Clinical Manifestations Related to Blepharospasm

At baseline, all the patients showed moderate to severe degrees (Jankovic grading ≥ 3) in the severity and frequency of blepharospasm. Most of them reported functional blindness and had trouble performing daily life activities. No significant differences were observed between the BoNT-A group and the BoNT-A+APP group before treatment (*p* = 0.586 in severity and *p* = 0.452 in frequency). After treatment, the severity and frequency of blepharospasm improved perceptibly in both the BoNT-A group and the BoNT-A+APP group (Figure 2).

Figure 3 shows the typical manifestations of blepharospasm patients before and after BoNT-A injection or BoNT-A+APP combined treatment. The eyelid spasms were notably improved in the two groups, with longer palpebral fissures and fewer periocular wrinkles after treatment. Appendix A demonstrates the effectiveness after BoNT-A injection.

### 3.3. Clinical Manifestations Related to Dry Eye

For the symptoms and signs of dry eye at baseline, no significant differences were found in OSDI, FL, FBUT and SIT between the patients treated by BoNT-A and BoNT-A+APP. Before treatment, all patients in the two groups had severe dry eye symptoms based on the high OSDI scores (BoNT-A group: 82.48 ± 7.37, BoNT-A+APP group: 78.82 ± 4.60, *p* = 0.112). However, dry eye signs were relatively weaker according to the results of FL (BoNT-A group: 3.18 ± 1.01, BoNT-A+APP group: 3.50 ± 1.24, *p* = 0.466), FBUT (BoNT-A group: 1.71 ± 0.77, BoNT-A+APP group: 2.17 ± 0.58, *p* = 0.077) and SIT (BoNT-A group: 2.53 ± 0.99, BoNT-A+APP group: 3.17 ± 1.23, *p* = 0.153). In the BoNT-A group, OSDI (*p* = 0.000) and FL (*p* = 0.003) were apparently decreased after treatment, with FBUT (*p* = 0.000) and SIT (*p* = 0.001) significantly increased (Figure 4). This indicates that BoNT-A could effectively improve dry-eye-related symptoms and signs in blepharospasm patients. The improvements in these indexes were more obvious when BoNT-A was used together with APP. Compared with the baseline, the OSDI score declined to 29.86 ± 6.24 after the BoNT-A+APP treatment, showing the effective improvement of the combined management in alleviating dry eye symptoms (*p* = 0.000, Figure 4A). Moreover, FL significantly decreased (*p* = 0.000, Figure 4B), while FBUT (*p* = 0.015, Figure 4C) and SIT (*p* = 0.000, Figure 4D) distinctly increased, confirming the therapeutic effect of the BoNT-A+APP combined treatment. As shown in Figure 3E,F, the corneal epithelial defect was obviously restored after the BoNT-A+APP treatment. Only one patient in each group had the complication of tearing, and there were no other side effects, such as punctal infection, ptosis and hypophasis.

Furthermore, to determine whether the combined management of BoNT-A and APP have a joint effect on the improvement of dry eye in blepharospasm patients, we compared the changing rates of OSDI, FL, FBUT and SIT between the BoNT-A group and the BoNT-A+APP group after 4–6 weeks of treatment. Compared with the BoNT-A group, evident improvements were observed in the reduction rates of OSDI (BoNT-A group: −52.23% ± 15.57%, BoNT-A+APP group: −61.84% ± 9.10%, *p* = 0.047, Figure 5A) and FL (BoNT-A group: −22.55% ± 25.98%, BoNT-A+APP group: −41.94% ± 14.46%, *p* = 0.016, Figure 5B) in the BoNT-A+APP group. But no significant differences were found in FBUT (BoNT-A group: 63.73% ± 63.53%, BoNT-A+APP group: 62.50% ± 89.37%, *p* = 0.968, Figure 5C) and SIT (BoNT-A group: 56.95% ± 56.05%, BoNT-A+APP group: 82.85% ± 63.72%, *p* = 0.270, Figure 5D) between the two groups.

## 4. Discussion

Clinically, blepharospasm patients have often been found to have dry eye manifestations, typically with a Schirmer test lower than 5 mm in 60% to 80% of cases [6,10]. Regarding the treatment, BoNT-A has been the primary therapy for blepharospasm and APP has shown a positive effect on the improvement of dry eye [22,23]. However, the combining treatment of BoNT-A and APP in patients with both blepharospasm and dry eye has remained elusive. In this investigation, we compared the effectiveness of BoNT-A plus APP treatment versus using BoNT-A alone on blepharospasm patients with dry eye.

In this study, we found a gender bias as 64.5% of patients in the BoNT-A group and 75% in the BoNT-A+APP group were female. This bias was consistent with previous studies. According to a retrospective study in Taiwan, a male/female ratio of 1.71 was observed in the incidence of blepharospasm and females had a significantly higher mean annual incidence than males in all age groups [24]. Another study in Brazil showed that 75.2% were female among 125 blepharospasm patients [25]. Furthermore, female predominance was also found in the Italian and Korean populations [26,27]. According to previous observational investigations, a multitude of blepharospasm patients also suffer from dry eye to different degrees [10,11]. The involuntary movement of the eyelid retractors and protractors in blepharospasm patients might affect tear dynamics, thus contributing to the development of dry eye [8]. In this study, we observed severe dry eye symptoms in all blepharospasm patients. The OSDI scores in the patients were fairly high compared to those in most studies [28,29]. This is mainly ascribed to the severity of blepharospasm in these patients. In most patients, severe eyelid spasms led to functional blindness and, therefore, generated an extremely high score in OSDI, as OSDI consists of a lot of self-assessed symptoms, i.e., reading, driving, working and watching TV, which are highly related to visual functions [19]. These subjective symptoms can be affected by blepharospasm rather than by the dry eye itself. As a result, the OSDI scores in this study could not represent the severity of dry eye in these blepharospasm patients. This was also confirmed through the relatively weak dry eye signs in the patients. Therefore, using OSDI in the diagnosis of dry eye might not be applicable to blepharospasm patients.

BoNT-A has been used to manage a wide variety of illnesses, including blepharospasm, strabismus, nystagmus, protective ptosis, cervical dystonia, eyelid retraction, hyperhidrosis, cerebral palsy and chronic migraine [30,31,32]. BoNT-A reduces muscle fiber activity by inhibiting the spreading of acetylcholine neurotransmitters at presynaptic nerve terminals by intermeddling with vesicle fusion [33,34]. This neurotoxin inhibits the release of local nociceptive neuropeptides and may moderate neurogenic inflammation and peripheral sensitization [34,35,36]. Several other treatment modalities, such as ocular and systemic medications and surgical procedures, have also been recommended in the treatment of blepharospasm [37]. Surgical options are limited to the rare patients that fail to respond to BoNT-A injection [1,38]. In previous studies, BoNT-A was shown to be effective and safe for the long-term treatment of blepharospasm. It has been proved as the treatment of choice for blepharospasm only if preparation, injection doses and techniques are chosen correctly [8,39,40]. The pretarsal orbicularis oculi muscle injection has been reported as the key component [39]. In our study, we confirmed the improvement of blepharospasm in all patients following BoNT-A injection.

Previous studies have confirmed the effect of BoNT-A injections in improving dry eye symptoms [1,15]. The medial lower pretarsal orbicularis oculi injections alleviate dry eye symptoms, although they are not indicated if the patient complains of watery eyes before treatment [39,41]. Lu et al. found that the mean of FBUT was significantly increased after BoNT-A injection and this treatment could effectively relieve dry eye symptoms in these patients and ameliorate their ocular surface condition [6]. Consistent with these studies, our study demonstrated significant differences in mean OSDI, FL, FBUT and SIT after BoNT-A injection, proving the pretarsal orbicularis oculi injections of BoNT-A had a positive effect in the treatment of dry eye in blepharospasm patients.

The punctal plug has been tested as a relatively effective, safe and reversible method of preserving artificial tears and the aqueous layer on the ocular surface in treating dry eye [42]. An APP may be inserted temporarily to assess the effectiveness of this therapy [43,44]. According to a prior study, APP improved dry eye visual function with a similar efficacy to that of punctate keratopathy and the same advantage in relieving ocular discomfort in dry eyes compared with artificial tears [41]. Plugs appear to have the best outcomes in maintaining the tear film by increasing tear volume. Additionally, it has a lasting influence on visual acuity, compared to the transient effectiveness of topical eye drops [41]. A punctal occlusion blocks the lacrimal outflow system at the punctum level and aims to maintain the tears produced naturally in the aqueous-deficient dry eye as well as to extend the contact time of artificial tears [42]. Dry eye patients treated with APP showed improvements in tear dynamics, dry eye symptoms, ocular surface health and visual acuity, and therefore indirectly enhanced compliance. Alfawaz et al. found that the percentage of normal eyes was higher in eyes with punctal plugs in terms of punctate epithelial keratitis score, FBUT and SIT compared to eyes without occlusion [43]. As it has been reported, patients with moderate to severe aqueous-deficient dry eye who do not respond to ocular medications are recommended for APP occlusion [16,42,43]. Furthermore, punctal plugs should be avoided in dry eye patients with nasolacrimal drainage system infections or from inflammatory causes [45].

Based on the positive effect of BoNT-A and APP on curing dry eye, we expected a joint effect of the combination of BoNT-A and APP in alleviating the dry eye symptoms in blepharospasm patients. Our study showed that BoNT-A plus APP insertion is better than only using BoNT-A in treating dry eye among blepharospasm patients. Compared with the BoNT-A group, the reductions in OSDI and FL in the BoNT-A+APP group were found to be more significant at the last follow-up visit, suggesting a superior improvement following the combined treatment of BoNT-A and APP. Indeed, our results prove the clinical efficacy of BoNT-A plus APP in the management of dry eye in blepharospasm patients. This treatment modality not only helped to restore the functional impairment of blepharospasm, including the severity and frequency of eyelid spasms, but also to reduce dry eye symptoms and corneal fluorescein staining. Therefore, we recommend a combination therapy of BoNT-A and APP in blepharospasm patients with severe dry eye symptoms, especially those who are found to have fluorescein staining in cornea.

## 5. Conclusions

In conclusion, this study suggests that OSDI is not applicable in the diagnosis of dry eye among blepharospasm patients. For blepharospasm patients with severe dry eye symptoms, especially those with fluorescein staining in cornea, the combined treatment of BoNT-A and APP is shown to be more effective compared with using BoNT-A alone. This study provides a feasible treatment option for blepharospasm patients with dry eye.

## Figures and Tables

**Figure 1 jcm-12-00877-f001:**
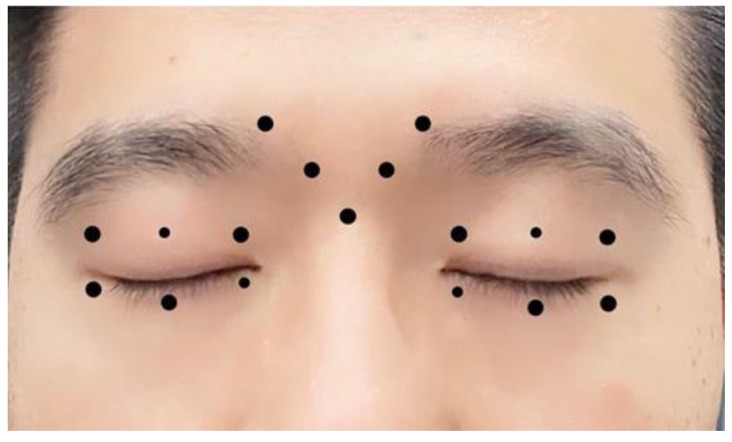
The sketch map of BoNT-A injection sites in blepharospasm patients with dry eye. Big dots represent 4 units and small dots mean 2 units of dose injected.

**Figure 2 jcm-12-00877-f002:**
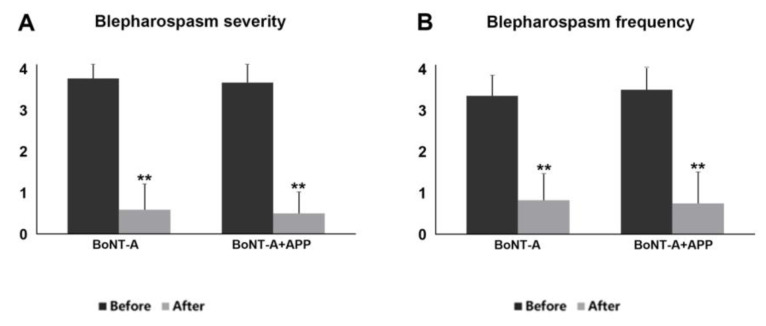
The (**A**) severity and (**B**) frequency of blepharospasm in the BoNT-A group and the BoNT-A+APP group before and after treatment (** *p* < 0.01).

**Figure 3 jcm-12-00877-f003:**
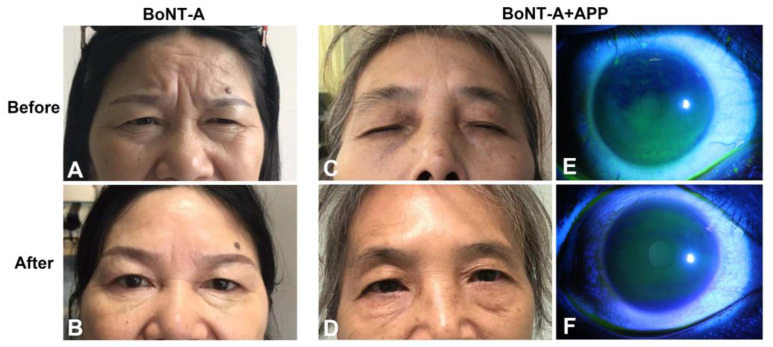
Typical manifestations of blepharospasm patients before and after treatment. (**A**,**B**) The facial characteristics before and after BoNT-A injection. (**C**,**D**) The facial characteristics before and after the combined treatment of BoNT-A and APP. (**E**,**F**) Corneal fluorescein staining before and after BoNT-A+APP combined treatment.

**Figure 4 jcm-12-00877-f004:**
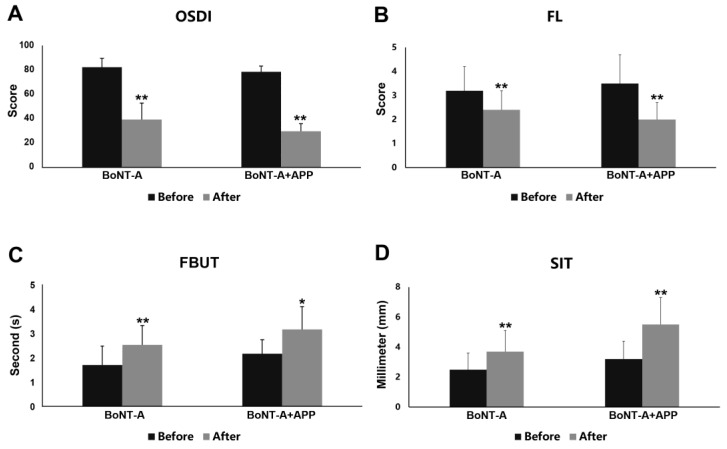
Dry eye symptoms and signs before and after treatment in the BoNT-A group and the BoNT-A+APP group. (**A**) OSDI score, (**B**) FL score, (**C**) FBUT (s) and (**D**) SIT (mm/5min). * *p* < 0.05, ** *p* < 0.01.

**Figure 5 jcm-12-00877-f005:**
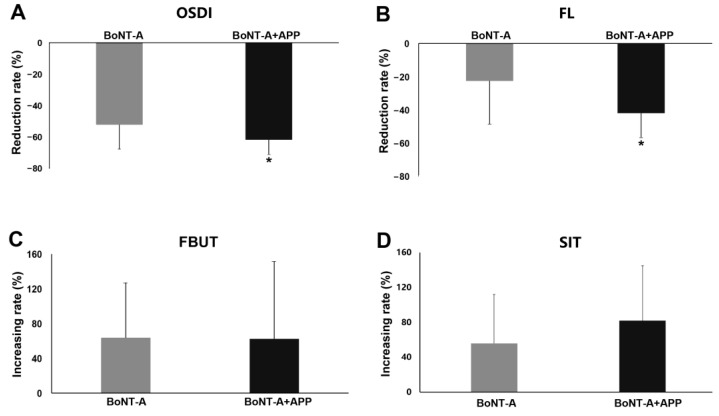
Changing rates (%) of dry eye symptoms and signs in the BoNT-A group and the BoNT-A+APP group before and after treatment. (**A**) OSDI, (**B**) FL, (**C**) FBUT and (**D**) SIT. * *p* < 0.05.

**Table 1 jcm-12-00877-t001:** Demographic characteristics of the blepharospasm patients with dry eye.

Characteristics	BoNT-A	BoNT-A+APP	*p* Value
N (%)	17 (100.0)	12 (100.0)	
Record eyes			
Right, n (%)	17 (100.0)	12 (100.0)	>0.999
Left, n (%)	0 (0.00)	0 (0.00)	
Gender			
Male, n (%)	6 (35.3)	3 (25.0)	0.555
Female, n (%)	11 (64.5)	9 (75.0)	
Age (years)			
Median (IQR)	67 (58.5–73.0)	64 (59.3–67.0)	
≤44, n (%)	1 (5.9)	1 (8.3)	0.406
45–59, n (%)	3 (17.6)	2 (16.7)	
≥60, n (%)	13 (76.5)	9 (75.0)	

BoNT-A: botulinum neurotoxin type A; APP: absorbable punctal plug.

## Data Availability

The datasets used and/or analyzed during the current study are available from the corresponding author upon reasonable request.

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
