# Peer review of "Botulinum Neurotoxin Type a Injection Combined with Absorbable Punctal Plug Insertion: An Effective Therapy for Blepharospasm Patients with Dry Eye"

_jcm, 2023, doi:10.3390/jcm12030877_

Round 1
Reviewer 1 Report
Interested paper. Although this is a novel investigation and the conclusions could be predicted, it is reasonable to formally study the question. Although the paper has limitations such as OSDI in the diagnosis of dry eye might not be applicable to blepharospasm patients, the authors do a good job of highlighting them.
Author Response
Thanks for your comments. Clinically, we found OSDI not suitable for the diagnosis of dry eye in blepharospasm patients. In this study, all the blepharospasm patients had high OSDI scores but relatively low degrees of FL, FBUT and SIT. This is probably because OSDI is based on subjective symptoms that may be caused by the blepharospasm rather than by the dry eye itself. This indicates that OSDI needs to be carefully evaluated regarding the severity of dry eye in blepharospasm patients.
Reviewer 2 Report
The manuscript is well written, well designed , the aim correlates with the title and methods, Results were scientifically well analyzed. Discussion is deep and conclusions are to the point
The idea of injection of Botulinum toxin and punctal plugs at the same time is worth to try in patients with blepharospasm The analysis of the data with improvement of the fluorescein break up time test approves the idea of adding value of the punctal plugs with release of the blepharospasm at the same time Of course it will be more convincing if the author had a control group of botulinum toxin only without punctal plugs to compare the results. The english language is acceptableAuthor Response
Thank you very much for your consideration of our manuscript. Our results showed that the combined treatment of botulinum toxin and punctal plug is an effective method for blepharospasm patients with severe dry eye symptoms, especially those with corneal fluorescein staining. Although we didn’t have a control group of using punctal plug only, we would add it in future research.
Reviewer 3 Report
Dear authors:
The work is exciting although I have some suggestions to make:
1. The sample size is too small
2. There is a gender bias since most patients are women.
3. The methodology should clarify whether the patients receive any treatment for the dry eye before and after surgery and whether it is the same in all cases.
4. As the authors clarify in the discussion and the conclusions, it seems that the OSDI is not very suitable for this type of blepharospasm patients as it is based on subjective symptoms of the patient that may be caused by the blepharospasm rather than by the dry eye itself.
5. There is no classification of patients according to the severity of their dry eye, so it is impossible to conclude whether they improve differently according to their severity.
Author Response
Thanks for your consideration of this manuscript. Your comments are all valuable and very helpful for revising and improving our paper. We have studied the comments carefully and have made corrections correspondingly. The revised portion is marked as “tracked changes” in the paper. The point-by-point responses are as following:
1. The sample size is too small
Response: Thank you for pointing out this issue. This paper does have the limitation of a small sample size. But we only enrolled 29 patients that were eligible for this study from January 1, 2020 until May 31, 2021. We will try to recruit more patients in future studies.
2. There is a gender bias since most patients are women.
Response: Yes, there is a gender bias in our study. But according to a reported study, females have a significantly higher mean annual incidence of blepharospasm than males in all age groups (https://www.ncbi.nlm.nih.gov/pmc/articles/PMC6306223/). Another study showed that 75.2% were female among 125 blepharospasm patients (https://pubmed.ncbi.nlm.nih.gov/22990720/). Female predominance of blepharospasm was also found in the Italian and Korean populations (https://pubmed.ncbi.nlm.nih.gov/16960861/;https://pubmed.ncbi.nlm.nih.gov/30311455/). We’ve added it in Discussion, lines 214-220.
3. The methodology should clarify whether the patients receive any treatment for the dry eye before and after surgery and whether it is the same in all cases.
Response: Thanks for your suggestion. All the patients were given standardized dry eye treatment using artificial tears (sodium hyaluronate eye drops; Santen Pharmaceutical Co., Ltd, Japan) three times a day before, but did not show any improvement in the dry eye symptoms. We’ve added the statement in lines 72-75.
After BoNT-A or BoNT-A+APP management, there was no specific treatment in those patients except using sodium hyaluronate eye drops three times a day if the patient needed it. We’ve supplemented it in lines 122-124.
4. As the authors clarify in the discussion and the conclusions, it seems that the OSDI is not very suitable for this type of blepharospasm patients as it is based on subjective symptoms of the patient that may be caused by the blepharospasm rather than by the dry eye itself.
Response: Thanks for your comments. That's exactly what we found in this study. In this study, all the blepharospasm patients had high OSDI scores but relatively low degrees of FL, FBUT and SIT. The results showed that OSDI might not be suitable for the diagnosis of dry eye in blepharospasm patients. This could be used as a reminder to ophthalmologists that OSDI needs to be carefully assessed regarding the severity of dry eye in blepharospasm patients as OSDI is based on subjective symptoms which can be affected by blepharospasm rather than by the dry eye itself. We’ve discussed this in lines 230-235.
5. There is no classification of patients according to the severity of their dry eye, so it is impossible to conclude whether they improve differently according to their severity.
Response: Thanks for pointing out this limitation. Although we didn’t classify the severity of dry eye in these patients, we did find significant improvements in corneal fluorescein staining, fluorescein tear break-up time and Schirmer I test after treatment (Figure 4). And we analyzed it in Results (lines 174-183) and Discussion (lines 256-259).
